# Research Progress in Porcine Reproductive and Respiratory Syndrome Virus–Host Protein Interactions

**DOI:** 10.3390/ani12111381

**Published:** 2022-05-27

**Authors:** Hang Zhang, Huiyang Sha, Limei Qin, Nina Wang, Weili Kong, Liangzong Huang, Mengmeng Zhao

**Affiliations:** 1School of Life Science and Engineering, Foshan University, Foshan 528000, China; hangzh2022@163.com (H.Z.); huiyangsha2022@163.com (H.S.); qlm@fosu.edu.cn (L.Q.); wang_nina@163.com (N.W.); 2Veterinary Teaching Hospital, Foshan University, Foshan 528000, China; 3Gladstone Institutes of Virology and Immunology, University of California, San Francisco, CA 94158, USA; weili.kong@gladstone.ucsf.edu

**Keywords:** porcine reproductive and respiratory syndrome virus, host protein, viral pathogenesis, interaction, research progress

## Abstract

**Simple Summary:**

PRRSV–host protein interactions are of great significance in finding new antiviral targets. We summarize the effects of all the PRRSV structural and nonstructural protein interactions with host proteins and on the protein interactions of the virus itself. These results provide a theoretical foundation for the therapeutic targeting of the intrinsic immune factors of the host protein and exploring viral replication mechanisms.

**Abstract:**

Porcine reproductive and respiratory syndrome (PRRS) is a highly contagious disease caused by porcine reproductive and respiratory syndrome virus (PRRSV), which has been regarded as a persistent challenge for the pig industry in many countries. PRRSV is internalized into host cells by the interaction between PRRSV proteins and cellular receptors. When the virus invades the cells, the host antiviral immune system is quickly activated to suppress the replication of the viruses. To retain fitness and host adaptation, various viruses have evolved multiple elegant strategies to manipulate the host machine and circumvent against the host antiviral responses. Therefore, identification of virus–host interactions is critical for understanding the host defense against viral infections and the pathogenesis of the viral infectious diseases. Most viruses, including PRRSV, interact with host proteins during infection. On the one hand, such interaction promotes the virus from escaping the host immune system to complete its replication. On the other hand, the interactions regulate the host cell immune response to inhibit viral infections. As common antiviral drugs become increasingly inefficient under the pressure of viral selectivity, therapeutic agents targeting the intrinsic immune factors of the host protein are more promising because the host protein has a lower probability of mutation under drug-mediated selective pressure. This review elaborates on the virus–host interactions during PRRSV infection to summarize the pathogenic mechanisms of PRRSV, and we hope this can provide insights for designing effective vaccines or drugs to prevent and control the spread of PRRS.

## 1. Introduction

Porcine reproductive and respiratory syndrome (PRRS) is a disease with a high incidence due to sow miscarriage, stillbirth, and piglet respiratory tract infections. PRRS is caused by porcine reproductive and respiratory syndrome virus (PRRSV) [1,2,3]. The disease was first reported in the United States in 1987 and then spread worldwide [4,5]. China reported the first case of PRRS in 1996. In 2006, there was an outbreak of a “high fever symptom” caused by a mutant PRRSV with a 30-amino-acid deletion in the nonstructural protein (NSP) 2 [6]. PRRSV continues to mutate, and a series of new PRRSV mutant strains such as the NADC30-like strain have appeared in recent years [7,8], making the prevention and control of the disease increasingly difficult. PRRSV continues to recombine and mutant, and since PRRSV has the antibody-dependent enhancement effect, the development of effective vaccines against PRRS is not as good as expected.

PRRSV is a single-stranded positive-strand RNA virus with an envelope belonging to the *Arterivirus* family. Other viruses in the same genus include equine arteritis virus (EAV), murine lactate dehydrogenase virus (LDV), and monkey hemorrhagic fever virus (SHFV) [3]. Pigs are the only naturally infected hosts of PRRSV. PRRSV directly infects porcine alveolar macrophages (PAMs) in pigs. In vitro, some passaged cell lines such as MARC-145 and MA-104 can also be infected by PRRSV [9].

The PRRSV genome is 15 kb in length and contains 10 open reading frames (ORFs), with ORF1a, ORF1b, ORF2a, ORF2b, ORF3, ORF4, ORF5, ORF5a, ORF6, and ORF7 from the 5′ end to the 3′ end, respectively. Each ORF partially overlaps with its adjacent ORF. ORF1a and ORF1b mainly encode viral replicases, accounting for approximately 80% of the viral genome. ORF2-5 encode the proteins GP2a, GP2b, GP3, GP4, GP5, and GP5a, respectively; ORF6 encodes the matrix protein (M); and ORF7 encodes the viral nucleocapsid protein (N) [10]. ORF1, encoding a replicase, can be further hydrolyzed into 16 nonstructural proteins (NSP1α, NSP1β, NSP2-6, NSP-2N, NSP-2TF, NSP7α, NSP7β, and NSP8-12) [11]. This review focuses on recent advances to understand the interactions between PRRSV proteins and host cellular factors (Figure 1), which can lay the foundation for finding new antiviral targets and exploring the mechanism of viral replication.

## 2. Nonstructural Proteins

### 2.1. NSP1

PRRSV NSP1 is the first protein produced after virus invasion, and there are two papain-like cysteine proteinase (PCP) active regions on NSP1α and NSP1β. After PRRSV synthesizes the polymer protein PP1a in the host cell, the NSP1α and NSP1β at the top of PP1a are dissociated by cutting the polyprotein via their own PCP activity [12]. NSP1 is involved in the processing, transcription, and regulation of host cell innate immunity genes, and is a multifunctional regulatory protein. NSP1 interacts with a variety of cellular proteins to regulate the PRRSV replication; these proteins are shown below.

Cholesterol-25-hydroxylase (CH25H) is a conserved interferon-stimulated gene (ISG) encoding endoplasmic reticulum-associated proteases that catalyze cholesterol production into 25HC. NSP1α was reported to interact with CH25H, which degrades NSP1α through the ubiquitin–proteasome pathway. This interaction inhibits PRRSV replication. The K169 site in the NSP1α protein is a key site for ubiquitination [13]. The TRAF-interacting protein (TRAIP) is an E3 ubiquitin ligase that plays an important role in the immune response. NSP1α interacts with K205, the locus of the TRAIP, and NSP1α reduces SUMOylation and K48 ubiquitination. The regulation of this double modification of the TRAIP by NSP1α leads to over-enrichment of the TRAIP in the cytoplasm, which in turn leads to excess K48 ubiquitination and degradation of the serine/threonine protein kinase (TBK1), thereby antagonizing TBK1–IRF3–IFN signaling and promoting PRRSV proliferation [14]. Linear ubiquitination is a newly discovered post-translational modification catalyzed by a linear ubiquitin chain assembly complex (LUBAC). NSP1α inhibits the LUBAC-mediated activation of NF-κB; the CTE domain is required for this inhibition. Mechanistically, NSP1α binds to HOP/HOIL-1L and impairs the interaction between the HOIL-1-interacting protein (HOIP) and SHARPIN. This reduces the LUBAC-dependent linear ubiquitination of the NF-κB essential modulator (NEMO). In addition, the PRRSV infection blocks formation of the LUBAC complex and NEMO linear ubiquitination, thereby facilitating PRRSV replication [15]. NSP1α also interacts with the host E3 ubiquitin ligase ankyrin repeat and SOCS box-containing 8 (ASB8). Specifically, porcine ASB8 induces the ubiquitination of K63 junctions with the stability of NSP1α to promote PRRSV replication. In addition, ASB8 is phosphorylated by the host IκB kinase β (IKKβ) at the N-terminal Ser-31 residue. In turn, ASB8 promotes the ubiquitination of the K48-linked residue and the degradation of IKKβ through the ubiquitin–proteasome pathway, significantly inhibiting I-kappa-B-α (IκBα) and p65 phosphorylation, thereby inhibiting NF-κB activity [16].

In addition to the interaction of NSP1α with CH25H, Dong et al. [17] found that the His-159 residue in NSP1β plays a key role in the reduction of CH25H. They further showed that NSP1β mediates the degradation of CH25H in HEK293FT cells via the lysosomal pathway, and that the anti-PRRSV activity of CH25H can be inhibited by NSP1β in MARC-145 cells. Using co-immunoprecipitation (Co-IP) experiments, Βeura et al. [18] confirmed that the cellular poly(C)-binding proteins 1 and 2 (PCΒP1 and PCΒP2) colocalize to the replication–transcriptional complex (RTC) of the virus and interact with NSP1β in regulating PRRSV, demonstrating that this interaction plays an important role in viral RNA composition. Another study showed that the interaction between residues of NSP1β at amino acids 85–203 (PCPβe and CTE domains) and residues of PCΒP2 at amino acids 96–168 (KH2 domain) is favorable for the replication of highly pathogenic PPRSV (HP-PRRSV) in MARC-145 cells [19]. NSP1β blocks the nuclear translocation of interferon-stimulating gene 3 (ISGF3) by inducing the degradation of karyopherin-alpha1 (KPNA1). Moreover, valine-19 in NSP1β is related to inhibition [20]. It has also been reported that NSP1β inhibits the expression of ISG15 and ISG56, and blocks the nuclear translocation of signal transducer and activator of transcription 1 (STAT1), thereby inhibiting interferon (IFN) signaling [21]. NSP1β binds to the cellular protein nucleoporin 62 (Nup62), causing the nuclear pore complex (NPC) to be decomposed, blocking the nuclear cytoplasmic transport of host mRNA and host proteins, and ultimately inhibiting host antiviral protein expression and viral effects on host translation mechanisms [22].

Using a tumor necrosis factor-alpha (TNFα) promoter system, NSP1 was found to strongly inhibit TNFα promoter activity. This inhibition occurred at the proximal region of the promoter. Both NSP1α and NSP1β inhibited TNFα promoter activity. Furthermore, a transcription factor (TF)-specific reporter plasmid was found to bind to TNFα promoters. It was demonstrated that NSP1α and NSP1β bind to CRE-κB(3) and Sp1, respectively, to transactivate elements that inhibit the activity of TFs. Subsequent analysis showed that NSP1 partially inhibited NF-κΒ activation, and NSP1β completely abolished Sp1 transactivation. These findings revealed an important mechanism of innate immune evasion by PRRSV [23].

### 2.2. NSP2

NSP2 is the largest and most variable nonstructural protein of PRRSV, ranging from the N-terminus to the C-terminus and including the protease region (PL2), hypervariable region, transmembrane region, and cysteine-rich conserved region [24]. NSP2 plays an important role in viral replication, and PRRSV accomplishes its own viral replication by interacting with host cell proteins.

DEAD-box RNA helicase 18 (DDX18) is a member of the DEAD-box RNA helicase (DDXs) family and can be involved in viral replication. Jin et al. [25] demonstrated that NSP2 overexpression in MARC-145 cells and primary PAMs enables the transfer of DDX18 from the nucleus to the cytoplasm; NSP2 recruits DDX18 into the viral replication complex to enhance PRRSV replication. Interleukin-2 enhancer-binding factor 2 (ILF2) is involved in many cellular pathways and is involved in the life cycle of some viruses. Wen et al. [26] discovered that cellular ILF2 can participate in a variety of cell pathways and some parts of the viral life cycle, as it interacts with NSP2 in vitro and plays a negative regulatory role in PRRSV replication. Song et al. [27] used Co-IP and stable isotope labeling with amino acids in cell culture (SILAC) to identify NSP2-interacting proteins during PRRSV infection. They found that NSP2 interacted with the known receptor of PRRSV, vimentin, to form complexes that may be essential for PRRSV attachment and replication, revealing a certain role of NSP2 in PRRSV replication and immune escape. Using quantitative label-free proteomics, Xiao et al. [28] identified that the NSP2 hypervariable region interacts with 14-3-3 proteins. More specifically, the subtype 14-3-3 epsilon was shown to interact with NSP2, which plays a role in the replication of HP-PRRSV TA-12 strains [29]. A recent study showed that 14-3-3 epsilon also plays an important role in NSP2-induced autophagy by binding to the tail domain of NSP2 [30]. Han et al. [31] found that NSP2 exists in different isomer forms during PRRSV infection. Heat shock 70 kDa protein 5 (HSPA5), as a cell chaperone associated with NSP2, is important for PRRSV replication. The triggering receptor expressed on myeloid cells 2 (TREM2) acts as an anti-inflammatory receptor that negatively regulates the innate immune response. Zhu et al. [32] found that NSP2 interacts with TREM2 to promote PRRSV infection. TREM2 downregulation leads to early activation of P13K/NF-κB signaling, thus reinforcing the expression of proinflammatory cytokines and IFN-I. Due to the enhanced cytokine expression, a disintegrin and metalloproteinase 17 was activated to promote the cleavage of membrane CD163, which resulted in suppression of infection. The NSP2 protein PL2 domain plays an important role in the proteolysis of PRRSV replication enzymes. PL2 belongs to the superfamily of deubiquitinating enzymes (DUΒs) in the ovarian tumor domain, which inhibits the generation of ISG15 and inhibition of ISG15 coupling with cellular proteins. ISG15 and ISGylation play an important role in PRRSV infection, and NSP2 interacts with ISG15 to counteract its antiviral function [33].

### 2.3. NSP4

PRRSV NSP4 is a 3C-like serine protease (3CLSP) that cleaves NSP3 into NSP12 [34]. NSP4 is a catalytic triplet of the conserved amino acids His39–Asp64–Ser118 and is the main protease involved in the expression and processing of viral proteins.

Swine ribonuclease L (sRNase L) is an antiviral protein that is induced by IFNs. Through experiments with luciferase activity assays, Zhang et al. [35] found that the PRRSV proteins that interact with sRNase L are NSP4, NSP12, and N, which are colocalized within cells. Yin et al. [36] screened host proteins that interact with NSP4 in PAM cDNA libraries and found that, as a new negative regulator in the innate immune response, porcine immunoglobulin lambda-like peptide 5 (sIGLL5) interacts with NSP4, and they are colocalized in HEK-293 cells. During PRRSV infection, NSP4 can mediate the partial transfer of TRIM28 from the nucleus to the cytoplasm by interacting with TRIM28, a member of the TRIM family, in a nuclear transcription protein CRM1-dependent manner. The nucleation of TRIM28 acts as an E3 ubiquitin ligase, which causes the early autophagy regulator Vps34 to undergo ubiquitin-like protein modification, enhances the interaction between Vps34 and the key regulatory protein complex of autophagy Beclin1, and regulates the formation of autophagosomes in cells [37]. This study thus revealed a new molecular mechanism by which PRRSV induces autophagy, elucidating the interaction mechanism between PRRSV and host cells from the perspective of protein post-translational modification. RNA-binding motif protein 39 (RBM39) is a transcriptional coactivator of AP-1/Jun, an estrogen receptor, and NF-kappa B, and involves precursor mRNA splicing. RBM39 was shown to promote PRRSV infection by interacting with NSP4 to regulate c-Jun phosphorylation [38]. In addition, NSP4 can interact with F-actin and myosin IIA to complete the transmission and infection of PRRSV between cells [39].

### 2.4. NSP5

The structure and function of the PRRSV NSP5 protein remain unclear. Signal transducer and activator of transcription 3 (STAT3) is a pluripotent signaling mediator of many cytokines, including interleukin (IL)-6 and IL-10. STAT3 plays a key role in cell growth, proliferation, differentiation, immunity, and the inflammatory response. Different strains of PRRSV-1 and PRRSV-2 were infected with MARC-145 and primary PAMs. All infections resulted in a significant decrease in STAT3 protein levels in a dose-dependent manner. NSP5 induces STAT3 degradation by increasing its polyubiquitination levels and shortening its half-life from 24 h to 3.5 h. The C-terminal domain of NSP5 is the key region for inducing the degradation of STAT3. In addition, STAT3 signaling was significantly inhibited in cells transfected with NSP5-expressing plasmids. These results suggest that PRRSV NSP5 antagonizes STAT3 signaling by accelerating STAT3 degradation through the ubiquitin–proteasome pathway [40].

### 2.5. NSP7

The PRRSV NSP7 protein consists of an internal cleavage site that can be further cleaved into two proteins, NSP7α and NSP7β [41]. NSP7 plays a role in stimulating the humoral immune system in PRRSV-infected pigs, but its structure and function are still not fully understood. Chen et al. [42] analyzed the expression of NSP7α and NSP7β in PRRSV-infected MARC-145 cells and compared the NSP7α of EAV and the NSP7α of PRRSV using nuclear magnetic resonance, showing that both proteins have three α-helices clustered together and the β-sheets of both proteins are located on one side of the α-helices. Helix α2 to helix α3 was the region showing the strongest overlap between the two proteins. Although this structural analysis provided little insight into protein function, based on the structure of NSP7α, some key amino acids on NSP7α (such as F72) have been identified to interact with NSP9, suggesting that NSP7α may be found in the form of an RTC and could assist the role of NSP9 in PRRSV RNA synthesis. Further research is needed to verify whether NSP7α interacts with NSP9 and if it participates in viral replication [42].

### 2.6. NSP9

PRRSV NSP9 is an RNA-dependent RNA polymerase (RdRp), which plays a vital role in viral replication. DDX5 is a cellular protein that interacts with NSP9, and the two proteins were found to colocalize in the cytoplasm by a yeast two-hybrid (Y2H) screen of the PAM cDNA library. Further studies demonstrated that the DEXDc and HELIC domains of DDX5 interact with the RdRp domains of NSP9. In PRRSV-infected MARC-145 and PAMs, endogenous DDX5 showed co-positioning with NSP9 [43]. NSP9 of PRRSV-2 colocalizes with the cell retinoblastoma protein (pRb) in the PRRSV-infected MARC-145 and PAM cytoplasm, and NSP9 promotes pRb degradation through the proteasome pathway. The interaction of NSP9 with pRb facilitates the reproduction of PRRSV-2 in vitro [44]. Dong et al. [45] utilized Y2H, Co-IP, GST pull-down, and immunofluorescence assays to show that full-length Annexin A2 (ANXA2) can interact with NSP9 in vitro, and NSP9 in PRRSV-infected MARC-145 cells interacts with endogenous ANXA2. Nucleotide-binding oligomerization domain-like receptor (NLR) X1 is unique among NLR proteins, and acts as an antiviral factor for different viral infections. PRRSV infection promotes expression of the NLRX1 gene, which in turn inhibits PRRSV replication in MARC-145 cells, whereas knockdown of NLRX1 enhances PRRSV infection in PAMs. Mechanically, NLRX1 impairs the accumulation of intracellular, viral subgenomic RNA (sgmRNA). Mutagenesis analysis showed that the leucine-rich repeat sequence (LRR) domain of NLRX1 interacts with the RdRp domain of NSP9 to exert antiviral activity [46]. Wen et al. [26] used Co-IP to analyze the interaction between ILF2 and NSP9 in 293FT and MARC-145 cells, and the colocalization of ILF2 and NSP9 was detected by confocal immunofluorescence. They further found that the RdRp domain of NSP9 interacted with ILF2. This interaction causes ILF2 to translocate from the cell nucleus to the cytoplasm along with NSP9 in PRRSV-infected MARC-145 and PAMs; the knockdown of ILF2 favors PRRSV replication, whereas the overexpression of ILF2 inhibits PRRSV replication in MARC-145 cells [26]. Using transcriptome sequencing, Zhao et al. [47] discovered that a CCCH-type zinc finger protein, the zinc finger antiviral protein (ZAP), expression was upregulated in MARC-145 cells transfected with MAVS and ZAP. This upregulation inhibited PRRSV infection during the early stages of replication. NSP9 interacts with ZAP, and the location of the interaction was mapped to the zinc finger domain of ZAP and the N terminal of NSP9 amino acids 150–160. These findings suggest that ZAP is an effective antiviral factor that inhibits PRRSV infection, providing new insight into virus–host interactions [47].

### 2.7. NSP10

PRRSV NSP10 is a helicase with a thermolabile and pH-sensitive NTPase activity consisting of a zinc finger motif at the N-terminus and a superfamily 1 (SF1) domain at the C-terminus helicase [48]. NSP10 plays a vital role in viral replication, and as one of the most conserved proteins of PRRSV, is a good candidate as a diagnostic marker [49]. Jin et al. [25] discovered the interaction between DDX18 and NSP10 through Co-IP experiments, and positioned the binding region of DDX18 at the N- and C-termini of NSP10. The expression of NSP10 in MARC-145 and primary PAMs enables DDX18 to redistribute from the nucleus to the cytoplasm, which then promotes viral replication. Silencing of the DDX18 gene in MARC-145 cells suppressed PRRSV replication. These findings demonstrate that DDX18 plays an important role in PRRSV replication, providing insights into the replication of PRRSV [25].

### 2.8. NSP11

PRRSV NSP11 is a nidovirus-specific endoribonuclease (NendoU) with uridine properties that has a catalytic effect on the entire NendoU family with complete conservation of 162-bit His, 178-bit His, and 224-bit Lys. Mutations or deletions of amino acids in the NendoU structure can affect viral replication, transcription, and the production of infectious viruses [50,51]. Shi et al. [52] demonstrated that PRRSV NSP11 promotes PRRSV infection in MARC-145 cells. The endoribonuclease activity of NSP11, rather than its deubiquitinating activity, is critical for p21 degradation. NSP11 mediates p21 protein degradation via ubiquitin-independent proteasome degradation [53]. Wang et al. [54] found that NSP11 relies on NendoU to inhibit the production of IFN-I, inhibit ISRE promoter activity, and facilitate ISG transcription. Detailed analysis showed that NSP11 overexpresses the C-terminal correlation domain targeting IRF9 to inhibit IFN signaling, rather than STAT1 or STAT2. In addition, the NSP11–IRF9 interaction impairs the formation and nuclear translocation of ISGF3 in the context of NSP11 overexpression in PRRSV-infected cells [54]. CH25H is an important ISG-encoded multi-bit membrane protein that significantly inhibits the replication of many viruses. Amino acids such as His-129, His-144, and Lys-173 in NSP11 were identified to play a key role in the reduction of CH25H. In addition, NSP11 mediates the degradation of CH25H via the lysosomal pathway in HEK293FT cells. The anti-PRRSV activity of CH25H can be antagonized by NSP11 in MARC-145 cells [17]. The linear ubiquitination-specific deubiquitinase ovarian tumor domain deubiquitinase with linear linkage specificity (OTULIN) can control the immune signaling transduction pathway by restricting the Met1-linked ubiquitination process. The interaction between porcine OTULIN and NSP11 was found to be dependent on the OTU domain. NSP11 promotes the degradation of NEMO linear ubiquitination by recruiting OTULIN, resulting in a superposition effect that inhibits the production of IFN-I [55]. Tripartite motif-containing 59 (TRIM59) was cotransfected with NSP11 in HEK293T and PRRSV-infected PAMs, and the interaction domain was identified as the N-terminal ring domain in TRIM59 and the C-terminal NendoU domain in NSP11. In addition, the overexpression of TRIM59 could inhibit PRRSV infection in MARC-145 cells. Conversely, transfection with TRIM59 small interfering RNA (siRNA) leads to an increase in the production of PRRSV in PAMs. This indicates that the interaction of TRIM59 with NSP11 may inhibit PRRSV replication [56]. NSP11 overexpression inhibited IFN generation by NSP11-dependent RLR pathways with double-stranded RNA analogues, and TNF-α was found to interact with MAVS and RIG-I to suppress IRF-3 and NF-κΒ activity, respectively [57]. Li et al. [58] used a Y2H system to prove that galectin-1 (Gal-1) interacts with the NSP11 NendoU domain by protein imprinting and viral titer assays, which indicated that Gal-1 overexpression inhibits PRRSV proliferation. The N-terminal domain (NTD) of NSP11 is responsible for STAT2 degradation and interacts with the STAT2 NTD and the coiled-coil domain. Mutagenesis analysis showed that the amino acid residue K59 of NSP11 is indispensable for inducing STAT2 reduction. In summary, NSP11 antagonizes IFN signaling by mediating STAT2 degradation, which provides further insights into the innate immune PRRSV interference mechanism [59].

### 2.9. NSP12

The structure and function of PRRSV NSP12 remain unknown. There are reports that the combination of cysteine 35 and cysteine 79 in NSP12 is required for sgmRNA synthesis [60]. To explore the function of NSP12, Dong et al. [61] studied the interaction of NSP12 with cell proteins using an immunoprecipitation strategy of a quantitative proteomics-binding NSP12-EGFP fusion protein overexpressed in 293T cells to determine whether HSP70 can interact with NSP12. They found that NSP12 recruits HSP70 to maintain its stability and inhibits viral replication [61]. Porcine galactoctin-3 (GAL3) is a 29 kDa protein encoded by a single gene, LGAS3, located on chromosome 1. Li et al. [62] demonstrated that GAL3 interacts with NSP12 and directly inhibits PRRSV replication. Karyopherin alpha 6 (KPNA6) is required for PRRSV replication, and Yang et al. [63] discovered that NSP12 can induce KPNA6 stabilization and KPNA6 silencing was not conducive to PRRSV replication.

### 2.10. NSP3, 6, and 8

The structure and function of PRRSV NSP3, NSP6, and NSP8 have not been reported, and information on their interactions with host proteins is limited. The intrinsic virus-limiting factor (IFITM1) inhibits infection with a wide variety of viruses. NSP3 interacts with the porcine IFITM1 distributed around the periphery of the nucleus, induces IFITM1 degradation in PRRSV during infection, and counteracts the antiviral function of IFITM1; thus, further analysis of this protein can provide new clues for exploring the mechanisms associated with PRRSV-evading host immune recognition [64]. Exostosin glycosyltransferase 1 (EXT1), an enzyme involved in the biosynthesis of heparin sulfate, has been reported to be a host factor essential for a wide variety of pathogens. Zhang et al. [65] found that EXT1 had no effects on the attachment, internalization, or release of PRRSV, but did restrict viral RNA replication. EXT1 was determined to interact with viral NSP3 via its N-terminal cytoplasmic tail and to enhance K48-linked polyubiquitination of NSP3 to promote their degradation.

## 3. Structural Proteins

### 3.1. GP2

The PRRSV-2 GP2a glycoprotein contains 256 amino acid residues and has a molecular weight of approximately 29 to 30 kDa; the structure includes a signal sequence of residues 1–40 at the N-terminal, an outer membrane domain composed of 168 residues, a transmembrane region, and 20 residues that comprise the intramembrane domain [66]. GP2a is a type I membrane integrin whose C-terminal is anchored to the membrane and combines with other structural proteins by disulfide bonds to form a multimer. The PRRSV-2 GP2 protein contains 171 or 178 amino acid residues, with two conserved glycosylation sites; however, the glycosylation sites are not necessary for viral infection [67]. The GP2a protein has poor immunogenicity and is found low in the PRRSV virion.

GP2a acts as a viral attachment protein responsible for mediating interactions with CD163 to bring the virus into susceptible host cells. The C-terminal 223-bit residues of the CD163 molecule are necessary for susceptibility to PRRSV infection in BHK-21 cells, but are not required to interact with GP2a [68]. Gao et al. [69] found that GP2a targets the unfolded protein response (UPR). The central regulator GRP78 undergoes proteasomal degradation, whereas the UPR IRE1–XΒP1 and PERK–eIF2α–ATF4 signaling branches are both turned on in the early stages of infection. The activated effector XΒP1s enters the nucleus, and ATF4 is incidentally transferred to the cytoplasmic viral replication complex via NSP2/3 to promote viral RNA synthesis [69].

### 3.2. E

The PRRSV envelope (E) protein is a minor structural component of virions, which is important for viral infectivity and is encoded by ORF2b. Zhang et al. [70] used Co-IP and colocalization assays to confirm that tubulin-α is the interacting partner of the E protein. The binding of the C-terminal 25-bit residue of the E protein with tubulin-α is critical to their interaction. Overexpression of the E protein in cultured cells caused microtubule depolymerization [70]. Pujhari et al. [71] found that the E protein and mitochondria are part of the ATP synthase complex (ATP5A) interaction and induce apoptosis by inhibiting ATP production. The ubiquitin–proteasome pathway plays a major role in pCH25H degradation during PRRSV infection, where the E protein interacts with and degrades pCH25H by ubiquitination at the ubiquitination site pCH25H Lys28 [72]. Interaction of Gal-1 with the E protein inhibits the proliferation of PRRSV in PAMs and MARC-145 cells [58]. The E protein also interacts with tetherin and removes part of tetherin from the cell surface to inhibit its antiviral activity [64].

### 3.3. GP3

PRRSV-1 and PRRSV-2 GP3 proteins contain 265 and 254 amino acid residues, respectively. As a secondary protein for the capsule membrane, GP3 is a highly glycosylated protein with poor conservatism between strains, and the mutation site occurs mainly at the N-terminal [73,74]. Su et al. [75] identified PRRSV genetic variants that interacted with IFN-α via complete genome sequencing. In these mutations, an amino acid substitution of GP3 (F143L) was found to enhance IFN-α resistance by enhancing the inhibitory effect of pJAK1. Wang et al. [76] generated monoclonal antibodies to GP3, designated as 1F10, which could recognize a new epitope (68–76 amino acids). Li et al. [77,78] constructed a recombinant adenovirus expressing the HSP70 gene and HP-PRRSV GP3, and found that HSP70 could fuse with the GP3 of PRRSV. Significantly, this interaction induced the production of IFN-γ and IL-4 in porcine serum, enhanced the immune response effect, and conferred protection against PRRSV infection in pigs, providing new ideas for vaccine design [77,78].

### 3.4. GP4

The PRRSV-1 and PRRSV-2 GP4 proteins consist of 183 and 178 amino acid residues, respectively, and their structure consists of an N-terminal cleaved signal peptide of 1 to 21 residues and a transmembrane region of amino acid residues located between positions 156 and 177. GP4 contains four potential glycosylation sites (amino acid residues 37, 84, 120, and 130) [79]. GP2, GP3, and GP4 are thought to combine to form a polycomplex located on the surface of the capsule membrane [68]. PRRSV contains the main glycoprotein GP5 on the virion envelope, as well as three other secondary glycoproteins, namely, GP2a, GP3, and GP4, all of which are necessary for the production of infectious virions. GP4, together with GP2a, is essential for mediating interglycoprotein interactions and acts as a viral attachment protein responsible for mediating interactions with CD163 to bring PRRSV into susceptible host cells [68].

### 3.5. GP5a

In recent years, a novel structural protein, ORF5a, was found in all arteriviruses where it is encoded by an alternative ORF of the sgmRNA encoding the main envelope glycoprotein GP5, suggesting its important role in virology. Oh et al. [80] established a subline of PAMs to stably express the PRRSV ORF5a protein and assessed the expression levels in PAMs at different times by proteomic analysis. A total of 36 protein spots were found to be differentially expressed. The identified cellular proteins are involved in various cellular metabolism-related processes such as cell growth, cytoskeletal networks, cell communication, metabolism, protein biosynthesis, RNA treatment, and transport. These proteomics data thus will provide valuable information for gaining a better understanding of the specific cellular responses to the novel ORF5a protein during PRRSV replication [80].

### 3.6. GP5

PRRSV GP5, a major membrane protein and the most varied structural protein in PRRSV with a molecular weight of approximately 25 kDa, is essential for the assembly of viral particles and is involved in viral pathogenesis. MYH9 is the heavy chain of nonmuscle myosin IIA (NMHC-IIA), which is involved in internalization of the PRRSV virion. The MYH9 C-terminal domain (designated PRA) interacts with GP5, which triggers the assembly of PRA and endogenous MYH9, thereby inhibiting PRRSV infection [81,82]. Zhang et al. [83] confirmed the interaction between GP5 and ATP5A through Co-IP, indicating that GP5 plays an important role in regulating cellular ATP production. Su et al. [75] found that an amino acid substitution of GP5 (Y136H) enhances IFN-α resistance by promoting the inhibitory effect of pJAK1. GP5 and M proteins form heterodimer complexes, with great significance for viral structure and infectivity. Snapin is a coprotein of the SNARE membrane fusion network and a member of the BLOC-1 complex. Hicks et al. [84] used Y2H to find that both GP5 and M proteins can specifically interact with cell snapin, and play an important role in intracellular transport and membrane fusion. Similar to GP3, Li et al. [78,85] found that HSP70 fused with the GP5 of PRRSV, which had the same protective effects on virulence in terms of enhancing the cytokine production and the immune response.

### 3.7. M

The M protein is one of the main components of the PRRSV membrane protein, and is the most conserved and important structural protein of the virus. M and GP5 proteins combine with disulfide bonds to form heterodimers, accounting for almost half of the entire structural protein, which is of great significance for viral structure and infectivity [86,87,88]. Hicks et al. [84] used a Y2H screen to discover that both GP5 and M proteins can specifically interact with snapin in the host cell, and play a role in intracellular transport and membrane fusion. Wang et al. [89] used M monoclonal antibodies (mAbs) from MARC-145 cells infected with PRRSV HuN4-F112 to identify host cell proteins that interact with the M protein of HP-PRRSV. Two cellular proteins of interest were validated by Co-IP and confocal analysis: the nuclear factor of activated T cells 45 kDa (NF45) and proliferating cell nuclear antigen (PCNA) were found to interact with the M protein, which aids in future research aimed at exploring the role of the M protein in PRRSV replication and pathogenesis.

### 3.8. N

The N protein is a versatile, alkaline phosphate protein encoded by the ORF7 gene with a molecular weight of approximately 15 kDa. The N protein, which is the main antigen protein of PRRSV, is highly immunogenic. The main antigenic determinant of the N protein is the conformational epitope, which forms a β-fold of amino acids at its C-terminal and is an important structure for maintaining its antigenicity [90,91]. The N protein is associated with the viral RNA genome that plays a role in genome packaging, and the N protein itself can form a homologous dimer [92]. There is a nuclear localization signal (NLS) between amino acids at positions 41 and 47 of the N protein, which are highly conservative determinants. After porcine infection with PRRSV, the early immune response is mainly against the N protein, followed by the M and GP5 proteins; therefore, the N protein has important value in the serological diagnosis of PRRSV infection.

Song et al. [93] used a Y2H assay to identify HIC proteins as inhibitors of MyoD family-a (I-mfa) as the cell chaperone of the N protein. HIC is a zinc-binding protein, and confocal microscopy showed that N and HIC-p40 isomers colocalize in the nucleus and nucleoli, whereas HIC colocalizes with p32 in the cytoplasm; moreover, the interaction between N and cell transcription factors has a regulatory effect on host cell gene expression [93]. SUMOylation is a reversible post-translational modification that regulates the function of the target protein. The N protein can interact with only the SUMO E2-coupled enzyme Ubc9, which colocalizes with the N protein in the cytoplasm and nucleus. Overexpression of Ubc9 inhibits viral genomic replication in the early stages of PRRSV infection, and knockdown of Ubc9 by siRNA promotes viral replication. These findings revealed the SUMOylation properties of the N protein and the participation of Ubc9 in PRRSV replication by interacting with multiple proteins in PRRSV [94]. N interacts with the host cell’s RNA helicase DExD/H-box helicase 9 (DHX9) and redistributes proteins into the cytoplasm. Knockdown of DHX9 increases the proportion of short sgmRNA, whereas DHX9 overexpression favors the synthesis of longer sgmRNA and viral genomic RNA (gRNA). These results suggested that DHX9 is recruited by the N protein during PRRSV infection to regulate viral RNA synthesis [95]. Myxovirus resistance 2 (Mx2) was identified as a novel IFN-induced innate immune limiting factor that inhibits certain viral infections. Monkey Mx2 (mMx2) interacts with N proteins in virus-infected cells. The interaction mechanism suggests that the GTPase activity of mMx2 is necessary, but the first N-terminal amino acid at position 51 is dispensable for antiviral activity. Wang et al. [96] found that porcine Mx2 has antiviral activity against PRRSV in MARC-145 cells, suggesting that the mMx2 protein inhibits PRRSV replication through interaction with the viral N protein. The poly(A)-binding protein (PAΒP) is a host cell protein that improves translation efficiency by cyclic mRNA and was identified to interact with PRRSV by Y2H screening cell chaperones of the N protein. The interaction domain of the N protein and PAΒP was identified at the 52–69 amino acid region using a series of deletion mutants. PAΒP silencing in cells mediated by short hairpin RNA (shRNA) led to PRRSV RNA synthesis, viral-encoded protein expression, and a significant decrease in viral titer. Together, these results showed that the PAΒP plays an important role in regulating PRRSV replication [97]. Yoo et al. [98] demonstrated that the N protein colocalizes with small nucleolar RNA (snoRNA)-associated fibrin. The interaction domain of N and fibrin was identified at the 30–37 amino acid regions. For the first 80 amino acids of fibrin, the glycine-arginine-rich region (GAR domain) containing glycine-arginine-rich regions was identified as the domain that interacts with N [98]. TRIM25 is an inhibitor of PRRSV replication, and a Co-IP assay showed that the N protein interferes with the TRIM25–RIG-I interaction by competing with TRIM25. The N protein inhibits the expression of TRIM25 and TRIM25-mediated RIG-I ubiquitination to inhibit IFN-β production [99]. Liu et al. [100] used label-free quantitative proteomics to identify several cytokines that may interact with N proteins, including poly (ADP-ribose) polymerase 1 (PARP-1), which adds an adenosine diphosphate ribose to proteins. Treatment of PRRSV-infected cells with the PARP-1 small-molecule inhibitor 3-AΒ suggested that PARP-1 is necessary in PRRSV replication [99]. Sagong et al. [101] identified 23 cellular proteins that interact with N proteins, which are involved in various cellular biological processes such as cell division and metabolism, inflammatory response, stress response, the ubiquitin–proteasome pathway, protein folding and synthesis, and transport. It is worth noting that expression of the heat shock 27 kDa protein (HSP27) is upregulated in PAM-pCD163-N cells. Overexpression of monkey viperin (mViperin) was shown to inhibit PRRSV proliferation by blocking the early steps of PRRSV invasion, as well as viral genome replication and translation, but did not inhibit assembly and release. mViperin interacts with N proteins at different cellular plasma sites. In addition, it was found that amino acids 13–16 of mViperin are essential for inhibiting PRRSV replication [102]. The TRIM22 protein is one member of the TRIMs family that has been identified as a key limiting factor for inhibiting human viruses. Jing et al. [103] used Co-IP analysis to find that TRIM22 interacts with the NLS2 motif of the N protein through the SPRY domain. A new host limiter, Moloney leukemia virus 10-like protein (MOV10), was identified as an inhibitor of PRRSV replication. Co-IP and immunofluorescence colocalization analyses showed that MOV10 interacts with and colocalizes with N proteins in the cytoplasm. MOV10 affects the distribution of the N protein in the cytoplasm and nucleus, resulting in N protein retention in the former, and MOV10 inhibits PRRSV replication by restricting the nuclear input of the N protein [104]. DExD/H-box helicase 36 (DHX36) is an ATP-dependent RNA helicase that unravels guanine-quadruplex DNA or RNA, which plays the role of a pattern recognition receptor in the innate immune response. PRRSV infection increases the expression of DHX36. When myeloid differentiation primary response gene 88 (MYD88) and its adapter were knocked down by siRNA in MARC-145 cells, the activation of NF-κB and expression of pro-inflammatory cytokines were reduced after PRRSV infection. Further studies showed that the N protein interacts with the N-terminal quadruplex binding domain of DHX36 to enhance N protein-induced NF-κΒ activation [105]. S100A9 is a damage-related molecular pattern of the S100 protein family, which inhibits PRRSV replication in a cell in a Ca^2+^-dependent manner. The N protein colocalizes with S100A9 in the cytoplasm, and the proteins interact via amino acid residue 78 of S100A9 and N protein residues 36–37. S100A9 may limit PRRSV proliferation by interacting with the viral N protein [106]. Yu et al. [107] found that the CD163 SRCR5 domain colocalizes with the N protein in the early endosomes and is involved in viral uncoating. The PAMs of the CD163 SRCR5 domain were resistant to PRRSV infection.

## 4. Summaries

Since the PRRS outbreak in the 1980s, efforts have been made to find effective antiviral strategies and vaccines to control its damage to the pig industry. However, this has become a significant challenge for a variety of reasons. As common antiviral drugs become increasingly inefficient under the pressure of viral selectivity, therapeutic agents that target intrinsic immune factors in host proteins are more promising as we speculate that targeting cellular proteins lead to mutations in the host proteins. Therefore, there is an urgent need to find new antiviral mechanisms and, on this basis, to develop therapeutic drugs [108]. Both nonstructural and structural viral proteins are targets for the attenuation of PRRSV virulent strains [109]. NSP2, NSP3, NSP10, GP2, and GP5 have been identified as the main proteins involved in viral decay [109]. In particular, NSP2 and GP5 are the most variable PRRSV proteins, and mutations usually occur in highly variable regions of these two proteins [110]. PRRSV NSP4- or NSP9-specific nanoantibodies can inhibit PRRSV replication by blocking these nonstructural proteins, which are key components of the viral RTC and, therefore, interfere with viral genomic replication and transcription [111,112]. According to our summary of the PRRSV–host interaction network, there are some relatively conserved nonstructural proteins, such as NSP1, NSP4, NSP9, and NSP11, which play an important role in the life cycle of viral infection; thus, targeting nanoantibodies to these proteins would be a good strategy. Inhibitors provide a new perspective on antiviral strategies that may potentially resist cross-genotype PRRSV infection. NSP1α, NSP1β, NSP10, and NSP11 provide a structural basis for the study of biological functions and mechanisms of action [12,113,114,115,116]. For example, the C-terminal extension of NSP1β binds to a putative substrate binding site of the PCP domain, illustrating the role of the substrate binding pattern and providing a structural template for the design of NSP1 inhibitors with potential therapeutic value [114].

Many viral proteins such as NSP1α, NSP1β, NSP2, NSP4, NSP7, NSP10, and NSP11 play a vital role in suppressing the host’s innate immune response. These key proteins can be used as antiviral targets in attenuated vaccines. For example, NSP2 inhibits IFN-β production by blocking IRF3 phosphorylation [117]. These modified viral proteins exhibit significantly reduced abilities to evade host immune responses, laying out new possibilities for the development of improved attenuated virus vaccines against arteriviruses. In addition, PRRSV NSP11 uses its deubiquitin activity to inhibit NF-κB signaling [118]. Thus, selective mutations will greatly increase the host’s innate immune response without affecting endogenous RNA activity. These mutations can be effectively applied to the design of live attenuated PRRSV vaccines.

In addition, the identification of microRNAs targeting the PRRSV genome provides an alternative target for gene editing [119]. With the rapid development of CRISPR/Cas-based gene editing technology, it is no longer a difficult task to obtain gene-edited pigs [120]. Studies have shown that pigs lacking the CD163 cell receptor or cysteine-rich scavenger receptor domain 5 (SRCR5) of CD163 are resistant to both PRRSV-1 and PRRSV-2 infections [121,122,123]. It is worth noting that the crystal structure of the SRCR5 domain of CD163 consists of seven β chains and two α spirals, which extends our interest in the SRCR5 domain for gaining a better understanding of PRRSV intrusion mechanisms [124]. Arg561 in the CD163 long ring region is an important residue for PRRSV invasion, and may play a key role in the interaction of CD163 and PRRSV during viral infection, providing a target for drug design and gene editing.

Owing to its rapid mutation rate, immunosuppression, and persistent infection characteristics, there are currently no effective drugs and vaccines to control PRRSV; compounded by its high morbidity and mortality, PRRS has become one of the world’s greatest hazards in the pig industry. During infection, there is a wealth of information available for host cell–virus interactions, and it is clear that PRRSV has evolved to be equipped with various strategies to disrupt the host antiviral system and provide favorable conditions for its own survival. Viruses interact with the host to evade the host’s immune response and then replicate and spread in the host, which is key to the survival of the virus.

## 5. Perspectives

Host factors required for viral replication are ideal drug targets because they are less likely to mutate under drug-mediated selective pressure compared with viral proteins. Thus, virus–host interactions are a new way to identify target host factors and guide antiviral drug development. For the whole PRRSV genome, there are four NSPs with enzymatic activity, namely, NSP2 with the cysteine protease activity [24], NSP4 with 3CLSP activity [34], NSP9 with RdRp activity [43], and NSP11 with nidovirus-specific endoribonuclease activity [50,51]. This information provides good ideas for the development of target therapeutic drugs. Regarding the host proteins, TRIM25, as an E3 ubiquitin ligase, interacts with the PRRSV N protein to inhibit ubiquitination of RIG-I and thus inhibits viral replication [99]. MOV10 is an interferon-induced RNA helicase that mediates the anti-PRRSV activity by binding to N proteins that prevent N proteins from entering the nucleus [104]. CH25H is a conserved ISG-encoding endoplasmic reticulum-associated protease, and CH25H can interact with NSP1α to degrade NSP1α through the proteasome pathway to inhibit viral replication [17]. These proteins may be able to function as a host target protein in the development of potential therapeutic agents.

## 6. Conclusions

As reviewed in this article, viruses utilize cellular mechanisms through protein interactions, and conversely, this also causes host immune defenses to limit viral infection. Therefore, potential therapeutic targets for the PRRSV–host interface are promising. This review summarized the host cell proteins that interact with various PRRSV proteins that have been identified in recent years, which provides a theoretical basis for finding new antiviral targets and exploring viral replication mechanisms.

## Figures and Tables

**Figure 1 animals-12-01381-f001:**
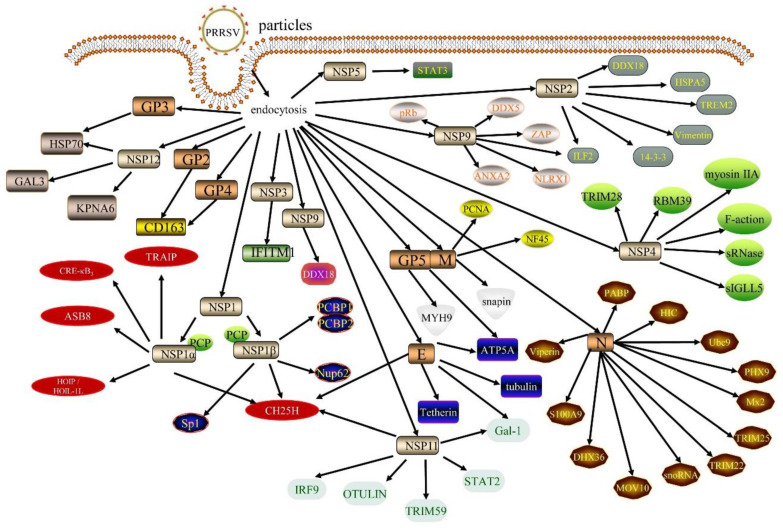
The interactions of the structural proteins of PRRSV and nonstructural proteins with host proteins mentioned in the article.

## Data Availability

All datasets are available in the main manuscript. The dataset supporting the conclusions of this article is included within the article.

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
