# Peer review of "Research Progress in Porcine Reproductive and Respiratory Syndrome Virus–Host Protein Interactions"

_animals, 2022, doi:10.3390/ani12111381_

Round 1
Reviewer 1 Report
This review was a relatively comprehensive packaging of the virus host interactions for PRRSV. I think this would be a great read for a somebody who is trying to learn about the molecular biology of PRRSV. This manuscript effectively distills the information from uniprot into a single document. There were however some interactions that are listed on uniprot that were not found in the manuscript. I would suggest the authors review uniprot for any interactions they may have missed.
A few minor fixes on line 54 it is unclear why ORF1a and ORF1b are mentioned explicitly while 2a and 2b, as well as 5a and 5b, are not explicitly mentioned.
Line 89 has a sentence that end in "... is required for inhibition." inhibition of what? I found this sentence confusing.
Thank you
Author Response
Dear reviewer 1:
We thank you for the constructive comments and suggestions to improve our manuscript. We agreed with most of the suggestions and comments. We have made modifications in the text to improve the manuscript quality. Our point-by-point responses are included as the following.
This manuscript effectively distills the information from uniprot into a single document. There were however some interactions that are listed on uniprot that were not found in the manuscript. I would suggest the authors review uniprot for any interactions they may have missed.
Revised. We have looked up the uniport and found the host proteins that interact with PRRSV contained in the database. They were already included in our manuscripts. The uniport database results show that NSP1α interacts with LUBAC, NSP2 interacts with DDX18, NSP3 interacts with IFITM1, NSP9 interacts with DDX5, NSP10 interacts with DDX18, NSP11 interacts with OTULIN, NSP12 interacts with LGALS3.
A few minor fixes on line 54 it is unclear why ORF1a and ORF1b are mentioned explicitly while 2a and 2b, as well as 5a and 5b, are not explicitly mentioned.
Revised. We have made the corresponding changes according to your requirements.
Line 89 has a sentence that end in "... is required for inhibition." inhibition of what? I found this sentence confusing.
Revised. “inhibition” means that NSP1α inhibits the LUBΑC-mediated activation of NF-κB.
Reviewer 2 Report
In the manuscript of Zhang et al., the authors review research progress in the area of PRRSV proteins – host protein interactions. PRRS is one of the most economically devastating diseases of pigs. Knowledge about viral – host protein interactions may contribute to the development of new antivirals.
Comments:
Line 37 – high mortality (20 – 100%) is a characteristic of highly pathogenic PRRSV strains that emerged in China in 2006. In general, PRRSV responsible for reproductive failure and respiratory problems in sows and piglets, respectively.
Line 42 – bad sentence. “… and then in 2006 there was an outbreak of highly pathogenic PRRSV (HP-PRRSV) characterized by high fiver and high pig mortality. HP-PRRSV strains contain 30-amino acid deletion in non-structural protein 2 (NSP2)”.
Line 65 – delete “This is”.
Line 89 – Replace “whose” with “which”.
Line 105 – Replace “experiments.” With “experiments,”
Line 336 – not clear meaning of “found in relatively few viruses.”
Line 337 and 338 – Put sentence in line 338 in front of the sentence that is in line 337.
Line 372 – Replace “virulence” with “infection”.
Line 383 – not clear meaning of “intermuscular protein”.
Line 387 – not clear meaning of “Arterioleitis viruses”. Do you mean arteriviruses?
Line 399 and 417 – Replace “capsule protein” with “membrane protein”.
Line 514-547 – Section 4 (Interaction proteomics) describes interaction between only viral proteins. The objective of the current review is to describe interactions between viral and host proteins. Therefore, I suggest to delete this section because it out of the scope of the review.
Line 557 – Replace “modifications” with “virulent strains”.
Line 559 – Replace “arterial viruses” with “arteriviruses”.
Line 587- Replace “for PRRSV” with “of CD163”.
In the manuscript of Zhang et al., the authors review research progress in the area of PRRSV proteins – host protein interactions. PRRS is one of the most economically devastating diseases of pigs. Knowledge about viral – host protein interactions may contribute to the development of new antivirals.
Comments:
Line 37 – high mortality (20 – 100%) is a characteristic of highly pathogenic PRRSV strains that emerged in China in 2006. In general, PRRSV responsible for reproductive failure and respiratory problems in sows and piglets, respectively.
Line 42 – bad sentence. “… and then in 2006 there was an outbreak of highly pathogenic PRRSV (HP-PRRSV) characterized by high fiver and high pig mortality. HP-PRRSV strains contain 30-amino acid deletion in non-structural protein 2 (NSP2)”.
Line 65 – delete “This is”.
Line 89 – Replace “whose” with “which”.
Line 105 – Replace “experiments.” With “experiments,”
Line 336 – not clear meaning of “found in relatively few viruses.”
Line 337 and 338 – Put sentence in line 338 in front of the sentence that is in line 337.
Line 372 – Replace “virulence” with “infection”.
Line 383 – not clear meaning of “intermuscular protein”.
Line 387 – not clear meaning of “Arterioleitis viruses”. Do you mean arteriviruses?
Line 399 and 417 – Replace “capsule protein” with “membrane protein”.
Line 514-547 – Section 4 (Interaction proteomics) describes interaction between only viral proteins. The objective of the current review is to describe interactions between viral and host proteins. Therefore, I suggest to delete this section because it out of the scope of the review.
Line 557 – Replace “modifications” with “virulent strains”.
Line 559 – Replace “arterial viruses” with “arteriviruses”.
Line 587- Replace “for PRRSV” with “of CD163”.
Author Response
Dear reviewer 2:
Thank you for the constructive comments and suggestions to improve our manuscript. We agreed to most of the suggestions and comments. We have made modifications in the text to improve the manuscript quality. Our point-by-point responses are included as the following.
Line 37 – high mortality (20 – 100%) is a characteristic of highly pathogenic PRRSV strains that emerged in China in 2006. In general, PRRSV responsible for reproductive failure and respiratory problems in sows and piglets, respectively.
Revised. We have replaced “high mortality” with “high incidence”
Line 42 – bad sentence. “… and then in 2006 there was an outbreak of highly pathogenic PRRSV (HP-PRRSV) characterized by high fiver and high pig mortality. HP-PRRSV strains contain 30-amino acid deletion in non-structural protein 2 (NSP2)”.
Revised. We have made the corresponding changes according to your requirements.
Line 65 – delete “This is”.
Revised. We have deleted “This is” as you suggest.
Line 89 – Replace “whose” with “which”.
Revised. We have replaced “whose” with “which” as you suggest.
Line 105 – Replace “experiments.” With “experiments,”
Revised. We have replaced “experiments.” with “experiments,” as you suggest.
Line 336 – not clear meaning of “found in relatively few viruses.”
Revised. We mean that GP2a protein has poor immunogenicity and is found less in PRRSV.
Line 337 and 338 – Put sentence in line 338 in front of the sentence that is in line 337.
Revised. We have made the corresponding changes according to your requirements.
Line 372 – Replace “virulence” with “infection”.
Revised. We have replaced “virulence” with “infection” as you suggest.
Line 383 – not clear meaning of “intermuscular protein”.
Revised. We have replaced “intermuscular protein” with “interglycoprotein”.
Line 387 – not clear meaning of “Arterioleitis viruses”. Do you mean arteriviruses?
Revised. We have replaced “Arterioleitis viruses” with “Arteriviruses”.
Line 399 and 417 – Replace “capsule protein” with “membrane protein”.
Revised. We have replaced “capsule protein” with “membrane protein” as you suggest.
Line 514-547 – Section 4 (Interaction proteomics) describes interaction between only viral proteins. The objective of the current review is to describe interactions between viral and host proteins. Therefore, I suggest to delete this section because it out of the scope of the review.
Revised. We have deleted the Section 4 (Interaction proteomics) according to your requirements.
Line 557 – Replace “modifications” with “virulent strains”.
Revised. We have replaced “modifications” with “virulent strains” as you suggest.
Line 559 – Replace “arterial viruses” with “arteriviruses”.
Revised. We have replaced “arterial viruses” with “arteriviruses” as you suggest.
Line 587- Replace “for PRRSV” with “of CD163”.
Revised. We have replaced “for PRRSV” with “of CD163” as you suggest.
Reviewer 3 Report
In this manuscript, Zhang et al., described advances in the Porcine Reproductive and Respiratory Syndrome virus (PRRSV)-host cell interactions research. The authors reviewed viral gene products and their association with cellular proteins including the functional significance of the interactions. Please consider the following comments before publication of the manuscript.
1. Most of the references used in the abstract are old. Recent publications PRRSV-host cell interactions should be included.
2. Describe why efforts to develop effective vaccines against PRRSV infection is not successful as it relates to virus-host interactions.
3. In some sections (for example: page 3 paragraph 1 and 2), there is lack of connection/flow which need to be corrected.
4. The Abstract is too superficial and needs to be improved.
5. Abstract lines 24 to 28: “……….this interaction activates the host cell immune response to clear the virus” is contradictory to some of the statements made in the body of the manuscript.
6. Lines 42 to 43: The sentence should be modified to “….a mutant PRRSV with 30 amino acid deletion in the non-structural protein (NSP)-2…….”
7. Fig 1. Was the figure developed by the authors or adopted from another publication? If not developed by the authors, please provide a reference.
8. Line 74: should read “host cell innate immunity genes”
9. Line 89 and line 114: “…….required for inhibition”. Inhibition of what?
10. Lines 131 to 134: Not clear what the authors try to convey. Please revise the paragraph!
11. Lines 139-140: Please describe how DDX18 facilitates the replication of PRRSV if known!
12. Lines 155-157: describe how the interaction of NSP2 with TREM2 promote PRRSV replication!
13. Line 168-169: the sentence is out of place.
14. Line 319: There are published articles which describe the interaction of NSP3 protein of PRRSV with cellular proteins. Those articles should be summarized under section 2.10
15. In a separate subheading, please summarize efforts that has been made to develop antiviral agents which target either viral or cellular proteins.
16. Section 5: Elaborate on potential side effects of targeting cellular proteins instead of viral proteins for the development of PRRSV antiviral drugs.
17. Lines 574 to 578 is not clearly described
18. Line 591: Arg561-in the long region of which protein
19. Lines 600 to 606: The conclusions made can be improved!
20. Based on the review, please suggest potential viral protein or cellular proteins as targets for antivirals
Author Response
Dear reviewer 3:
Thank you for the constructive comments and suggestions to improve our manuscript. We agree with most of the suggestions and comments. We have made modifications in the text to improve the manuscript quality. Our point-by-point responses are included as the following.
- Most of the references used in the abstract are old. Recent publications PRRSV-host cell interactions should be included.
Response: The recent publications on PRRSV-host cell interactions were included as you suggest. Line 339-345.
- Describe why efforts to develop effective vaccines against PRRSV infection is not successful as it relates to virus-host interactions.
Because PRRSV is constantly recombining and mutating, and PRRSV has the Antibody-Dependent Enhancement effect, the development of effective vaccines against PRRSV infection is not as good as expected. We think that from the perspective of virus-host interactions, it may provide new ideas for the development of new vaccines, because host proteins are not easy to mutate. These have been added in the line 52-54.
- In some sections (for example: page 3 paragraph 1 and 2), there is lack of connection/flow which need to be corrected.
Revised. We have made the corresponding changes according to your requirements. Line 83 to 84.
- The Abstract is too superficial and needs to be improved.
We have revised the Abstract as requested. Line 23 to 30 and line 36 to 39.
- Abstract lines 24 to 28: ……this interaction activates the host cell immune response to clear the virus” is contradictory to some of the statements made in the body of the manuscript.
Revised. We have made the corresponding changes according to your requirements. Line 33.
- Lines 42 to 43: The sentence should be modified to “……a mutant PRRSV with 30 amino acid deletion in the non-structural protein (NSP)-2…….”
Revised. We have made the corresponding changes according to your requirements. Line 48 to 50.
- Fig 1. Was the figure developed by the authors or adopted from another publication? If not developed by the authors, please provide a reference.
This figure was developed by the authors, and not cited from another publication.
- Line 74: should read “host cell innate immunity genes”
Revised. We have replaced “host cell innate immunity” with “host cell innate immunity genes” as you suggest. Line 83.
- Line 89 and line 114: “……required for inhibition”. Inhibition of what?
Line 98 “inhibition” means that NSP1α inhibits the LUBΑC-mediated activation of NF-κB.
Line 124 “inhibition” means that NSP1β blocks the nuclear translocation of interferon-stimulating gene 3 (ISGF3) by inducing the degradation of karyopherin-alpha1 (KPNΑ1).
- Lines 131 to 134: Not clear what the authors try to convey. Please revise the paragraph!
Revised. We have made the corresponding changes according to your requirements. Line 142 to 143.
- Lines 139-140: Please describe how DDX18 facilitates the replication of PRRSV if known!
Revised. We have made the corresponding changes according to your requirements. NSP2 recruits DDX18 into the viral replication complex to enhance PRRSV replication. Line 149 to 150.
- Lines 155-157: describe how the interaction of NSP2 with TREM2 promote PRRSV replication!
Revised. We have made the corresponding changes according to your requirements. TREM2 downregulation led to early activation of P13K/NF-κB signaling, thus reinforcing the expression of proinflammatory cytokines and IFN-I. Due to the enhanced cytokine expression, a disintegrin and metalloproteinase 17 was activated to promote the cleavage of membrane CD163, which resulted in suppression of infection. Line 169 to 172.
- Line 168-169: the sentence is out of place.
Revised. We have deleted this sentence.
- Line 319: There are published articles which describe the interaction of NSP3 protein of PRRSV with cellular proteins. Those articles should be summarized under section 2.10
Revised. The interaction of EXT1 and PRRSV NSP3 was summarized under section 2.10. Line 339 to 345.
- In a separate subheading, please summarize efforts that has been made to develop antiviral agents which target either viral or cellular proteins.
We have already added this paragraph: Line 592 to 607.
- Section 5: Elaborate on potential side effects of targeting cellular proteins instead of viral proteins for the development of PRRSV antiviral drugs.
Revised. We speculate that targeting cellular proteins may be led to mutations in the host proteins. Line 542 to 543.
- Lines 574 to 578 is not clearly described
Revised. We have made the corresponding changes according to your requirements. Line 565 to 567.
- Line 591: Arg561-in the long region of which protein.
Revised. Arg561-in the CD163 long region of which protein. Line 580.
- Lines 600 to 606: The conclusions made can be improved!
Revised. We have made the corresponding changes according to your requirements. Line 608 to 614.
- Based on the review, please suggest potential viral protein or cellular proteins as targets for antivirals.
Viral components: NSP9 and NSP11 have been reported as targets. The siRNA against them can inhibit viral replication.
Cellular protein components: TRIM25 and MOV10 could interact with PRRSV N protein to inhibit viral replication. CH25H could interact with PRRSV NSP1α to inhibit viral replication. Line 592 to 607. We feel that these are very promising targets.
